# Bilateral Transfer of Performance between Real and Non-Immersive Virtual Environments in Post-Stroke Individuals: A Cross-Sectional Study

**DOI:** 10.3390/ijerph20043301

**Published:** 2023-02-13

**Authors:** Deise M. S. Mota, Íbis A. P. Moraes, Denise C. R. Papa, Deborah C. G. L. Fernani, Caroline S. Almeida, Maria H. S. Tezza, Maria T. A. P. Dantas, Susi M. S. Fernandes, Alessandro H. N. Ré, Talita D. Silva, Carlos B. M. Monteiro

**Affiliations:** 1Physical Activity Sciences, School of Arts, Science and Humanities, University of São Paulo (EACH-USP), São Paulo 03828-000, Brazil; 2Rehabilitation Sciences, Faculty of Medicine, University of São Paulo (FMUSP), São Paulo 01246-903, Brazil; 3Faculty of Medicine, University City of Sao Paulo (UNICID), São Paulo 03071-000, Brazil; 4Medicine (Cardiology) at Escola Paulista de Medicina, Federal University of São Paulo (EPM/UNIFESP), São Paulo 04021-001, Brazil; 5Department of Physiotherapy, Universidade do Oeste Paulista (UNOESTE), Presidente Prudente 19050-920, Brazil; 6Department of Physiotherapy, University of Medical Sciences of Santa Casa of São Paulo, São Paulo 01224-001, Brazil; 7Department of Physiotherapy, Mackenzie Presbyterian University (UPM), São Paulo 01302-907, Brazil

**Keywords:** stroke, virtual reality, motor skills, motor activity

## Abstract

(1) Background: Post-stroke presents motor function deficits, and one interesting possibility for practicing skills is the concept of bilateral transfer. Additionally, there is evidence that the use of virtual reality is beneficial in improving upper limb function. We aimed to evaluate the transfer of motor performance of post-stroke and control groups in two different environments (real and virtual), as well as bilateral transfer, by changing the practice between paretic and non-paretic upper limbs. (2) Methods: We used a coincident timing task with a virtual (Kinect) or a real device (touch screen) in post-stroke and control groups; both groups practiced with bilateral transference. (3) Results: Were included 136 participants, 82 post-stroke and 54 controls. The control group presented better performance during most parts of the protocol; however, it was more evident when compared with the post-stroke paretic upper limb. We found bilateral transference mainly in Practice 2, with the paretic upper limb using the real interface method (touch screen), but only after Practice 1 with the virtual interface (Kinect), using the non-paretic upper limb. (4) Conclusions: The task with the greatest motor and cognitive demand (virtual—Kinect) provided transfer into the real interface, and bilateral transfer was observed in individuals post-stroke. However, this is more strongly observed when the virtual task was performed using the non-paretic upper limb first.

## 1. Introduction

Stroke is classically characterized as a neurological deficit attributed to an acute focal injury of the central nervous system by a vascular cause, including cerebral infarction, intracerebral hemorrhage, and subarachnoid hemorrhage [1], and is one of the leading causes of death and disability worldwide [2]. The most difficult disability presented is motor function deficits that affect the individuals’ mobility [3,4], their daily life activities [5,6], their participation in society, and their odds of returning to professional activities [7,8]. All of these factors contribute to a low overall quality of life [9,10,11]. Considering the post-stroke deficits, rehabilitation training is the most effective way to reduce motor impairments [12], and different approaches have been used and studied all over the world, including new technology such as virtual reality (VR). According to Laver et al. [13], VR and interactive video gaming have emerged as recent treatment approaches in post-stroke rehabilitation with commercial gaming consoles in particular, being rapidly adopted in clinical settings.

Although VR presents some adverse symptomatology (specially immersive VR) such as nausea, dizziness, disorientation, fatigue, and instability [14], there is evidence that the use of VR and interactive video gaming are beneficial in improving upper limb function when used as an adjunct to usual care (increasing overall therapy time) or when compared with the same dose of conventional therapy [13]. According to Kiper et al. [15], there are benefits in using virtual environments for upper limb improvement, even compared with conventional rehabilitation, that indicate a great future for VR rehabilitation in post-stroke individuals. Recent studies found positive results with the use of VR in individuals post-stroke, such as improvement in upper limb motor ability (and motor activity log: quality of movement), sensory function, mobility and physical domains, instrumental activities of daily living, and quality of life [16,17]. Regardless of the feasibility and benefits from the use of VR for post-stroke individuals’ rehabilitation of their upper limbs [17], we are more interested in investigating and identifying how the knowledge of motor learning could be influenced by virtual and real tasks.

According to Lefebvre et al. [18], in the field of neurorehabilitation, motor learning has recently become the focus of a great deal of attention. Motor skill learning is particularly attractive since practice-induced improvement in sensorimotor performance support the development of new aptitudes (skills), which provide the flexibility to adapt to changing conditions. The use of motor learning principles for the rehabilitation of upper limbs is crucial to reduce the effect of disease and promote motor improvement in feasible clinical treatments [15,19,20].

In the light of motor learning in the rehabilitation of post-stroke individuals, one special method that has been used is the possibility to transfer a practiced skill using a non-paretic upper limb to the paretic limb after stroke. Although some investigated this concept in the early twentieth century [21], studying the phenomenon describing how the performance of a skill with one hand seemed to “teach” the same skill to the other hand, this still attracts some interest [22]. Hand-to-hand transfer was defined as “bilateral transfer” (BT) and is supported by the knowledge that practice of one part of the body in performing a skilled act increases the ability of the bilaterally symmetrical part in performing the same act [21]. The concept of BT has been studied in healthy individuals in the upper limb with different aims, such as to verify the interlimb transfer in asymmetry [23] and BT of motor skills from the preferred to the non-preferred hand and vice-versa [24]. This phenomenon has been studied in stroke individuals with positive results. Ausenda et al. [25] have found evidence that the BT of motor skills is present in post-stroke individuals, and that it improves the ability of the affected hand in performing a task. They investigated BT using a “nine hole peg test” and found some interesting results, e.g., the execution speed of the nine hole peg test with the paretic hand, after training the non-paretic hand, was on average more than 22% faster than the value recorded at baseline. Moreover, the work of Iosa et al. [26] investigated the motor strategies at the basis of sensorimotor learning in BT in post-stroke participants and concluded that learning was not due to a mere sequence repetition, but a strategy chosen on the basis of previous performance. Thus, the affected hand of individuals with stroke may benefit from sensorimotor learning with the un-affected hand. Attention, motivation, and the performance of repetitive movements are important factors to provide BT, which should be explored [22].

Considering the positive use of BT and the possibility to use non-immersive VR in post-stroke rehabilitation, we are particularly interested in figuring out if there is transfer of performance from practicing a task using the non-paretic upper limb to the paretic limb, using a real or virtual task. Thus, we organized a protocol with a coincident timing task that could be executed using a virtual device (Kinect) or a real device (touch screen) and divided the participants (post-stroke and control group participants) into two groups: one group practiced the task using the real device (touch screen) and the other group practiced the same task using the virtual device (Kinect). Moreover, both groups (post-stroke and control participants) practiced with bilateral transference—independent of the device, all groups were divided again into subgroups, considering the post-stroke participants’ functional upper limb (i.e., one subgroup started practicing the task using the paretic upper limb and practiced with the non-paretic upper limb immediately after; the other subgroup started practicing the task using the non-paretic upper limb and practiced with the paretic limb immediately after), and for the control group, the paretic upper limb was represented by the less functional upper limb and vice versa.

Thus, we hypothesized that: (1) the control group would present better performance than post-stroke participants in all protocol phases and all groups would present improved performance independent of the upper limb used (paretic or non-paretic); (2) all groups would present improved performance independent of the interface used (real or virtual); (3) our third and main hypothesis was that participants post-stroke would present improvement transfer between upper limbs (bilateral transference), and this improvement would be more evident when associated with the virtual reality task.

## 2. Materials and Methods

### 2.1. Participants

The individuals of the sample were recruited from a specialized service in rehabilitation in São Paulo, Brazil. The participants were distributed in a quasi-experimental design, i.e., each participant recruited was allocated into a group according to gender and age in order to maintain the homogeneity of the groups. The control group consisted of individuals without sensorimotor deficits or pathological conditions, matched for sex and age with the stroke group, in a convenience sampling recruited from a technological school for young and older people in the city of São Paulo.

The inclusion criteria were: the participant must have a medical diagnosis of post-stroke sequelae, all assessed 6 months post-stroke [27], the participant could not have undergone surgery or chemical neuromuscular blockade in the upper limbs in the 6 months prior to participation in the study, the participant could not present structured osteoarticular deformities and other comorbidities, such as disorders in cognitive function that would prevent comprehension or execution of the proposed training, and the participant had to have sufficient motor conditions to stay in a seated position and move both upper limbs (paretic and non-paretic) in a range of movement good enough to play the game. The comprehension was analyzed by a pre-test with 3 attempts to execute the task. Participants who dropped out of the protocol were excluded.

In total, 138 potential participants were invited to participate in this study, although 2 of them dropped out after inclusion. After the exclusion criteria, 136 participants were included, 82 post-stroke individuals and 54 control (without motor deficits).

### 2.2. Neurological, Functional, and Mobility Evaluation

The individuals were analyzed to characterize the group before starting the task with: Orpington Prognostic Scale (OPS) to assess stroke severity [28]; Fugl-Meyer Assessment (FMA) to evaluate motor function, balance, sensation, and joint function [29,30]; Mini-Mental State Examination (MMSE) to verify the cognitive state [31,32]; Timed Up and Go (TUG) to evaluate overall functional mobility [33]; and Upper Limb Motor Ability Test (AMAT) to quantify upper extremity functional limitation [34,35].

### 2.3. Instruments

All individuals were evaluated and practiced a coincident timing task using a software developed in the department of the School of Arts, Sciences and Humanities, EACH-USP (first version used in 2014 by Monteiro et al. [36] and recently by Monteiro et al. [37] and Crocetta et al. [38]). The coincident timing task is an interception task that involves synchronization between eye and hand movements, and requires a strategy to start, slow down, and “stop” movement, in addition to accuracy and precision [39,40].

The game consisted of the participant attempting to slide the hand avatar (virtual interface—Kinect) coinciding with the lighting of the final bubble on the runway (Figure 1), or touching the computer screen (real interface—touch screen). The game offers coincident timing task (Figure 1) which, when executed, displays on the computer screen 10 balls to be accessed (red light) in sequence until the last ball, which is considered the target (gray stripe). The coincident timing game aims to evaluate the difference in time between the participant’s execution of the response and the arrival of the object to the target location, overall temporal accuracy, and, hence, demonstrates a coincidence-anticipation timing skill.

The technology of the game can store a number and relevant information, from individual data information such as date of birth, sex, topography of diagnosis, and researcher name to data concerning the individual’s performance such as trial and error numbers whether we used the Kinect interface or touch screen. Moreover, the game includes information about the beginning and end of the game and ball drop speed.

### 2.4. Procedures

The participants were positioned comfortably in a chair adjusted according to size and needs, along with a footrest so that they were positioned properly to enable task execution. After being seated, the experimenter explained the task verbally and provided three demonstrations of how to perform the coincidence timing tasks [36]. The participants that used the touch screen were instructed to place one hand (i.e., the paretic hand or non-paretic hand, determined by groups) on the mousepad as the initial position. The Kinect group participants were positioned 1 m from the computer screen and instructed to place the corresponding hand on their lap.

Once the first ball turned on, the individual had to move their hand to either reach the target by using the touch screen (representing the real interface) or making a hitting gesture in front of the Kinect device (representing the virtual interface) exactly at the moment the bottom target ball turned on. Different sounds were provided as feedback for a hit or miss during acquisition, retention, and transfer, the range of error for a hit being -200 to 200ms. Runtime was recorded by the game in all experimental stages, and each participant performed all tasks in blocks.

### 2.5. Participant Groups and Protocol

In this study, a coincident timing task with a short-term motor learning protocol was used according to Monteiro et al. [36,37], Bezerra et al. [41], Quadrado et al. [42], and Moraes et al. [43] (Appendix A). This kind of protocol is focused to show performance improvement. Thus, to answer our first and second question about the transfer between upper limbs (bilateral transference) and improvement in performance independent of the upper limb used (paretic or non-paretic), individuals with stroke were divided into two subgroups considering the upper limb used to start the task: Stroke—NPF (n = 34), who performed the first practice (Practice 1) using the non-paretic upper limb, and afterwards practiced using the paretic upper limb (Practice 2); and Stroke—PF (n = 48), who performed the first practice (Practice 1) using the paretic upper limb, and afterwards practiced using the non-paretic upper limb (Practice 2). Considering the control group, we analyzed individuals without motor deficits to identify a normal pattern of movement (n = 54). They performed the first practice (Practice 1) using the dominant upper limb, and the second practice (Practice 2) using the non-dominant upper limb (Figure 2).

To answer our third question, whether all groups would improve performance independent of the interface used (real or virtual), the participants were randomly allocated in another two subgroups using two interfaces, real and virtual, generating 6 subgroups: Real Stroke NPF (n = 17), Real Stroke PF (n = 27), Real Control (n = 27), Virtual Stroke NPF (n = 17), Virtual Stroke PF (n = 21), and Virtual Control (n = 27). The practice protocol for all participants were 20 attempts at the Acquisition phase at a moderate speed, that is, 500 ms between the lights of each ball (divided into 4 blocks of 5 attempts each—A1 to A4); 5 attempts in the Retention (R) phase (after an interval of five minutes without contact with the task); and 5 more attempts in Transfer (T) phase, in which the participants changed the interactive device they used. According to Seidler (2010), the transfer phase is important to the generalization of learning and refers to the degree to which newly acquired skills can be produced with a new effector, in a new workspace, or under new modes of movement. Thus, in our protocol, we adopted the change in devices as the Transfer phase (i.e., those who practiced using Kinect “Virtual” changed to touch screen “Real”, and vice-versa). Hence, the participant performed a total of 60 trials, 30 with one upper limb and 30 with the other upper limb, depending on the group sequence (Stroke—NPF or Stroke—PF). The different groups are presented as follows: To better clarify groups and subgroups, we organized a flowchart with all protocols (Figure 2).

### 2.6. Data Analysis

For the sample characterization variables, we used ANOVA for age (3 groups), and the non-paired t-test was applied for the OPS, FMA, MMSE, TUG, lesion time, and AMAT, subdivided into functional skills (FS) and movement quality (MQ). In turn, for the categorical variables sex, type of stroke, and side of hemiparesis, we applied the chi-square test. As dependent variables, the time error was considered, that is, the difference between the time in which the touch or gesture was registered in milliseconds and the moment in which the sphere reached the target (arrival time), the measures being: Absolute Error (AE), which demonstrates movement accuracy, and Variable Error (VE), which identifies movement precision. The dependent variables were submitted to MANOVA with the factors: 2 [Group: Stroke, Control], by 2 [Interface: Real, Virtual] by 2 [Upper limb used: Paretic or Non-Paretic], by 2 [Sequence: Paretic upper limb first (Stroke—PF); Non-paretic upper limb first (Stroke—NPF)], by 2 [Blocks: Acquisition (A1-A4); Retention (R); Transfer (T)], with repeated measurements in the last two factors (Practice Order and Block). For the Block factor, separate comparisons were made for Acquisition (first acquisition block A1 versus second A2, third A3, and last A4), Retention (third acquisition block A3 versus retention block R), and Transfer (third acquisition block A3 versus transfer block T). Partial eta squared (ηp^2^) was reported to measure effect size and interpreted as small (effect size > 0.01), medium (effect size > 0.06), or large (effect size > 0.14) [44]. Post hoc comparisons were performed using the Tukey HSD test. Values of *p* < 0.05 were considered significant. The statistical package was the Statistical Package for the Social Sciences (SPSS; IBM, Chicago, IL, USA), version 26.0.

## 3. Results

Data were collected from July 2019 to March 2020. A total of 138 potential participants were invited to participate in this study and 2 participants dropped out after inclusion, so 136 participants were included (see Figure 2). Characterization of the sample is presented in Table 1. No adverse effects occurred during our study; our task was semi-immersive VR, so the negative effects of VR were controlled.

### 3.1. Acquisition—A1–A4

MANOVA found significant effects in Group (Wilks lambda = 0.729; F3, 126 = 15.6; *p* < 0.001; ηp^2^ = 0.27), Interface (Wilks lambda = 0.710; F3, 126 = 17.1; *p* < 0.001; ηp^2^ = 0.29), Sequence (Wilks lambda = 0.938; F3, 126 = 2.76; *p* = 0.045; ηp^2^ = 0.06), and Block (Wilks lambda = 0.929; F3, 126 = 3.20; *p* = 0.026; ηp^2^ = 0.07), and significant interactions between Sequence and Group (Wilks lambda = 0.924; F3, 126 = 3.46; *p* = 0.018; ηp^2^ = 0.08), Sequence and Upper Limb Used (Wilks lambda = 0.902; F3, 126 = 4.59; *p* = 0.004; ηp^2^ = 0.10), Sequence, Group, and Upper Limb Used (Wilks lambda = 0.929; F3, 126 = 3.23; *p* = 0.025; ηp^2^ = 0.07), Sequence by Block by Group (Wilks lambda = 0.892; F3, 126 = 5.06; *p* = 0.002; ηp^2^ = 0.11) and Sequence by Block by Group by Interface (Wilks lambda = 0.926; F3, 126 = 3.36; *p* = 0.021; ηp^2^ = 0.07). Separate ANOVAs will be presented in the following paragraphs.

#### 3.1.1. Absolute Error—AE

The absolute error is illustrated in Figure 3. Significant effects were found in Sequence (F1, 128 = 6.20, *p* = 0.014, ηp^2^ = 0.05), Group (F1, 128 = 39.4, *p* < 0.001, ηp^2^ = 0.24), and Interface (F1, 128 = 6.61, p = 0.012, ηp^2^ = 0.08), as well as interactions between Sequence and MSU (F1, 128 = 13.6, *p* < 0.001, ηp^2^ = 0.10), Sequence, Group, and Upper Limb Used (F1, 128 = 6.77, *p* = 0.010, ηp^2^ = 0.05), and Sequence, Block, and Group (F1, 128 = 4.28, *p* = 0.041, ηp^2^ = 0.03). These results have interesting meanings:

(1) In both interfaces, the group that started with the non-paretic side (Stroke—NPF) did not present a significant difference in performance in the following practice with the paretic member, which means that there was a transfer of learning when the practice was performed first with the non-paretic upper limb, while the group that started with the paretic side (Stroke—PF) showed a significant difference in the second practice with the non-paretic side, both in the virtual interface (*p* < 0.001) and real interface (*p* = 0.007). In addition, an analysis was performed between the performance of the paretic side in both practices (blue line in Practice 1 versus green line in Practice 2), and there was a significant difference (*p* = 0.049), meaning previous training with the non-paretic upper limb assists in the transfer in later performance to the paretic upper limb (Stroke—NPF), as these participants showed better performance than participants who started with the paretic upper limb without prior training (Stroke—PF).

(2) There was a significant difference in performance between paretic and non-paretic upper limb in Practice 1 (in both real and virtual interfaces), in which subjects who performed the task with the paretic upper limb presented worse performance than with the non-paretic upper limb regardless of the interface used; however, in Practice 2, there was no difference (Figure 3, Table 1). During the entire practice, there was a difference between stroke and control participants, except in blocks A2, A3, and A4 of Practice 2 with the virtual interface, as well as in differences between the virtual and real interfaces, in which the virtual interface presented greater absolute error in every acquisition than the real interface.

There were main effects in Blocks, with performance improvement from the first (A1) to the second Acquisition block (A2) (F1, 128 = 5.44, *p* = 0.021, ηp^2^ = 0.04) and from A1 to the third Acquisition block (A3) (F1, 128 = 6.28, *p* = 0.012, ηp^2^ = 0.05), with no difference between A1 and the last Acquisition block (A4) for both groups.

#### 3.1.2. Variable Error—VE

The variable error pattern is illustrated in Figure 4. Significant effects were found in Blocks when comparing A1 with A2 (F1, 128 = 10.3, *p* = 0.002, ηp^2^ = 0.08), A1 with A3 (F1, 128 = 6.43, *p* = 0.012, ηp^2^ = 0.05), and A1 with A4 (F1, 128 = 4.24, *p* = 0.041, ηp^2^ = 0.03), Group (F1, 128 = 14.8, *p* < 0.001, ηp^2^ = 0.10), and Interface (F1, 128 = 48.8, *p* < 0.001, ηp^2^ = 0.28), without significant interactions. These results show that, in general, there was an improvement in the VE from the first Acquisition block A1 to the second A2, third A3, and last block A4. In addition, the control group presented better VE than the stroke group, and in the real interface, the VE was lower when compared to the interface without physical contact.

### 3.2. Retention

Figure 3 also shows the AE during retention. No significant effects or interactions were found between the final acquisition block and the retention block (both in MANOVA and ANOVA), which generally shows that the performance acquired during practice was retained. However, although there was no significant interaction indicated by ANOVA, the post hoc test showed that in the real interface, there was no retention of the third Acquisition block A3 in the Retention block R in both sequences, other than in the Stroke—NPF (*p* = 0.020) group. ANOVA showed a main effect in groups both in the AE (F1, 128 = 35.4, *p* < 0.001, ηp^2^ = 0.22) and in the variable error (F1, 128 = 12.1, *p* = 0.001, ηp^2^ = 0.09), in which the group with stroke had worse AE and VE when compared to the control group.

#### 3.2.1. Transfer Changing Interaction Device

MANOVA found significant effects in Blocks (Wilks lambda = 0.426, F3,126 = 5, *p* < 0.001, ηp^2^ = 0.57) and Groups (Wilks lambda = 0.831, F3,126 = 8.55, *p* < 0.001, ηp^2^ = 0.17), and significant interactions between Blocks and Interface (Wilks lambda = 0.580, F3,126 = 30.4, *p* < 0.001, ηp^2^ = 0.42) and Block, Group, and Interface (Wilks lambda = 0.904, F3,126 = 4.43, *p* = 0.005, ηp^2^ = 0.10). Separate ANOVAs for each error (AE and VE) are described below.

#### 3.2.2. Absolute Error—AE

ANOVA confirmed the significant effect in Blocks (F1, 128 = 16.0, *p* < 0.001, ηp^2^ = 0.11) and interaction between Blocks and Interface (F1, 128 = 82.6, *p* < 0.001, ηp^2^ = 0.39). The post hoc test showed that the participants of both groups who performed the practice in the real interface showed an improvement in the performance of the third Transfer block A3 to the virtual interface (T); on the other hand, those who performed the practice in the real interface (A3) presented worse performance in the transfer to the virtual interface (T). The post hoc test showed that the participants in both groups who performed the practice in the virtual interface showed an improvement in the performance of the third Transfer block A3 to the real interface (T); on the other hand, those who performed the practice in the real interface (A3) showed a worse performance in the transfer to the virtual interface (T).

A main effect was also found in the Group (F1, 128 = 24.6, *p* < 0.001, ηp^2^ = 0.16); this result shows that the stroke group presented a higher AE than the control group. However, the post hoc tests revealed that the Stroke—NPF group did not show differences in relation to the control group both in Practice 1 and in Practice 2 (in which they performed the task with the paretic upper limb) in both interfaces, on the other hand, the Stroke—PF group showed differences in relation to the control group in Practice 1 in the two interfaces, while in Practice 2 (in which the task was performed with the non-paretic upper limb), there was a significant difference only in the virtual interface.

Comparisons were also made between the Transfer blocks (T) of Practice 1 and Practice 2, in order to find differences in the changing of interaction device with the paretic and non-paretic upper limb. A difference was only found in the virtual interface between the transfers of Practice 1 and 2 of the Stroke—PF group, that is, with the paretic upper limb, they presented worse performance in Practice 1. Finally, in order to assess whether the group that performed Practice 2 with the paretic upper limb (Stroke—NPF) at the virtual interface actually showed an important transfer of performance to the real interface (T), the performance of this group in Practice 2 in the virtual interface was compared with the first block of the acquisition of Practice 1 in the real interface, and there was a significant difference (Figure 3).

#### 3.2.3. Variable Error—VE

Regarding the VE, there was a significant effect only in Blocks (F1, 128 = 149.1, *p* < 0.001, ηp^2^ = 0.54), in which there was a significant increase in the VE from A3 to the Transfer block (T). In addition, a main effect in the Group remained present (F1, 128 = 6.71, *p* = 0.011, ηp^2^ = 0.05). This result shows that the stroke group had a higher VE when compared to the control group (Figure 4).

There was no correlation between performance improvement during practice regarding Age and the OPS scores, MMSE, FMA, TUG, AMAT HF, AMAT, QM, and lesion time.

## 4. Discussion

The present study aimed to evaluate the improvement in upper limb performance in post-stroke individuals during the execution of a non-immersive virtual reality task comparing the use of the same task practiced with a real interface (touch screen) and a virtual interface (Kinect). Moreover, we used the concept of “bilateral” transfer, and post-stroke individuals practiced both tasks using their paretic upper limb first, followed by non-paretic upper limb, and the inverse sequence (i.e., the other group started with the non-paretic upper limb and, after, practiced with paretic upper limb). The result partially agrees with our hypothesis and we will discuss our data as follows.

### 4.1. Differences between Group (Post-Stroke and Control Group) and Upper Limb Used

Our first hypothesis was that the control group should present better performance than post-stroke individuals in all protocol phases, and that they would improve their performance independent of the upper limb used; both were partially confirmed. Our result showed that the control group presented better performance during most of the protocol (even considering absolute and variable errors); however, it was more evident when compared with the post-stroke paretic upper limb (i.e., in some protocol phases, the use of the non-paretic upper limb in the post-stroke group had similar performance to the control group). These results are in agreement with several studies comparing individuals with post-stroke and control groups. Kwon, Shin, and Son [45] studied post-stroke individuals using a visuo-spatial tracking task, Subramaniam et al. [46] used an upper extremity stand-reaching, Coderre et al. [47] used robotic technology and a reaching task to examine sensorimotor impairments, Subramanian et al. [48] used a task to compare motor performance and movement patterns, and Hussain, Murphy, and Sunneragen [49] used a target-to-target pointing task. All those studies presented the worst performance in post-stroke individuals, even using the non-paretic upper limb.

The difference between post-stroke individuals and the control group can be justified considering that stroke frequently leads to permanent neurological impairment associated with significant weakness, and sensorimotor functions are impaired bilaterally and asymmetrically [50,51]. According to Lang et al. [52] and Sathian et al. [53], changes in muscle tone and limitations in the range of motion of the upper limb joints are present in post-stroke individuals, either due to contracture or sagging. These factors contribute to impaired motor control, with slower and less precise movements. Moreover, upper limb movements in post-stroke individuals are slower, less smooth, more segmented, and less accurate than in healthy individuals [49,54]. Although those studies report a better performance in the control group, our study showed an interesting first result in that, with practice, post-stroke participants presented a similar result to the control group; however, those improvements were found only in the non-paretic upper limb during some practice moments (i.e., blocks A2, A3, and A4 in the virtual interface with absolute error). This result is in agreement with some studies using technological tasks with stroke; it is described by Cameirão et al. [55], who used a VR task to analyze gaming parameters, Torriani-Pasin et al. [56], who evaluated a coincident timing task in stroke non-paretic upper limbs (Bassin device), and Moliterno et. al. [57], who organized a practice with a computer task. All these studies found that the non-paretic upper limb can be improved and present similar performance to the control group.

Although our results point to similar performance in the non-paretic upper limb when compared with the control group, it is important to emphasize that these values occurred mainly after training with the paretic upper limbs (see Figure 3). It can be considered evidence that practicing a task with the paretic upper limb improves the performance of the non-paretic upper limb in the same task practiced shortly thereafter. Thus, regardless of deficits in the non-paretic upper limb such as reduced accuracy, performance tasks, and strength and functional ability [58,59], previous training with a paretic upper limb favored an improvement in performance in the non-paretic upper limb, approaching that of the healthy control group.

### 4.2. Improvement in Performance Considering Interface (Virtual and Real)

Our second hypothesis, that all groups should improve performance during practice independent of the interface used (real or virtual) with retention of the task, was partially confirmed too. We found improvement in performance in variable error during all protocols (A1–A4 and Retention), with better performance during the real interface for both groups. Considering absolute error, the results showed that all participants improved performance from block A1 to A2 and A3, but not in A4 with retention of the task (after 5 min rest), mainly in all post-stroke participants (e.g., even considering paretic and non-paretic upper limb). This means that during practice, post-stroke participants’ performance improved, but during the first 15 trials, after this repetition, the performance worsened independent of the device used (real or virtual), but with retention after 5 min rest. It is likely that this laboratory task made the practice boring and fatiguing, and those were the reasons for the worsening in performance during practice. Thus, a more interactive, mutant, stimulating, dynamic, and fun virtual reality task could have provided more interaction and improvement in the performance environment. According to Levin, Weiss, and Keshner [60] a competitive stimulus such as scoring games and interactive [13,61,62] and motivating [63,64,65] tasks increase interaction and provide better improvement in performance.

Considering fatigue, according to Cirstea, Ptito, and Levin [66] and Knorr, Rice, and Garland [67], the worsening of performance can be explained by the possible muscle fatigue, leading to a sluggish motor response resulting from changes in the central nervous system and sensory–motor mechanisms. Fatigue is a common complaint after stroke [68], and post-stroke fatigue is a multidimensional motor–perceptive, emotional, and cognitive issue [69,70,71]. According to Yang et al. [72], who analyzed healthy individuals in a VR task and proposed a similar result in post-stroke individuals, the motor performance of the upper limb in a virtual environment correlated with cognitive fatigue, and they concluded that a state of high fatigue significantly affects the performance in a task coordinating the wrist in rehabilitation. Interesting results were found when comparing performance during real and virtual devices. We observed that in all phases of the protocol, there was better performance of the task in the real interface in the absolute and variable errors. This interface offers, in addition to the visual and auditory feedback present in the virtual interface, tactile sensory information, and it seemed to be more functional.

Thus, physical contact also generates tactile information that can be used to adapt movement to the environment. In the real interface, the participant interacts directly with the objects (contact interaction between participant finger and computer screen), focusing attention on the ecological validity of the environment and the extent to which one feels comfortable with the task [73]. Some studies using VR show results similar to ours, with better performance in the real interface in different populations: patients with cerebral palsy [36], Duchenne muscular dystrophy [42], amyotrophic lateral sclerosis [74], and autism spectrum disorder [43].

### 4.3. Improvement Performance Considering Bilateral Transference between Environments (Virtual and Real)

Our third and main hypothesis was that participants post-stroke would present improved transfer between upper limbs (bilateral transference), and this improvement would be more evident when associated with the virtual reality task. We can emphasize two interesting results that partially confirm our hypothesis and that can support the future use of VR tasks for rehabilitation:(a)Transference from paretic to non-paretic upper limb: we found bilateral transference in the group that started with the paretic upper limb, i.e., the group that practiced with the paretic upper limb first presented better performance when practicing with the non-paretic limb posteriorly, independently of interface (virtual or real). According to Land et al. [75], the training carried out with a limb in post-stroke individuals can have a positive effect on the performance of the same task during the use of the untrained limb, which further reinforces that the most important factor is its potential to be an effective therapeutic technique for rehabilitation of individuals with motor disabilities as stroke survivors.(b)Transference from non-paretic to paretic upper limb: another interesting result is that the group which started with the non-paretic upper limb presented bilateral transfer for the paretic upper limb, showing similar results between paretic and non-paretic. However, this result was confirmed only in the virtual interface (i.e., when participants practiced the virtual task first with the non-paretic upper limb, they transferred skill to the paretic upper limb). This result is very interesting (see Figure 3—most important result): after practicing the virtual task, the non-paretic upper limb presented the best result in the transfer to the real interface (Transfer, Practice 2). This can be reinforced by the comparison between the result of this transference with the acquisition in the first sequence with the paretic upper limb in the real interface (i.e., Transfer in Practice 2 versus A1 in Practice 1—real interface).

We can only speculate that practicing a more difficult task in a virtual interface promotes motor learning adaptation that can have a positive influence in the real interface afterward. According to Moraes et al. [43], a higher level of difficulty in tasks due to lower proprioceptive or tactile feedback, and the higher movement demand from the Kinect interface, may all promote different sensory processing patterns causing difficulty in performing the task. The use of bilateral transfer as a treatment option in stroke rehabilitation is a relatively new concept; therefore, limited research exists in the area. In view of the studies, we can observe that neural plasticity is amplified, and not decreased, after the stroke. Greater gains in strength and neural plasticity can be induced by the activation of the preserved neural network between limbs, without directly involving the target muscles. Individuals with chronic stroke have difficulty accessing and receiving conventional strength or locomotion training due to lack of muscle strength and coordination on the most affected side; thus, bilateral transfer provides an easy-to-apply training paradigm to increase strength and the role of the most affected members [76].

Although we found interesting results, we can point out some limitations of the present study: (1) Considering that we divided the participants into groups and subgroups, a higher number of subjects could provide a larger effect size and external validity. (2) To better characterize the participants, we should have used cognitive (MMSE) and functional mobility (TUG) in the control group. (3) Depression, anxiety, and mood assessments for both the control and experimental groups could have been used to provide additional information. (4) We could have applied a questionnaire about the level of education and previous experience with technology to both the control and experimental groups. (5) We used a laboratory specific motor task and other VR tasks, and exergames should be assessed in the future to analyze performance, engagement, and motivation. (6) We organized a short-term protocol with only 1 day of practice, and these results cannot be generalized as a training protocol. Considering the above limitations, we are organizing a prospective randomized crossover-controlled trial with different virtual games, and using specific sensorimotor and cognitive assessments for the experimental and control groups. This future study will contribute to a better understanding of the positive and negative influences of virtual reality training during the rehabilitation of post-stroke individuals.

## 5. Conclusions

We can conclude that the task with the greatest motor and cognitive demand (Virtual interface—Kinect) provided transfer to the real interface, and bilateral transfer is observed, present in individuals after chronic stroke, evidenced by the practice of VR in a coincident timing task, and strongly observed when the task is started with a non-paretic upper limb, which reinforces the applicability of using this limb as a facilitator of motor learning in these individuals.

## Figures and Tables

**Figure 1 ijerph-20-03301-f001:**
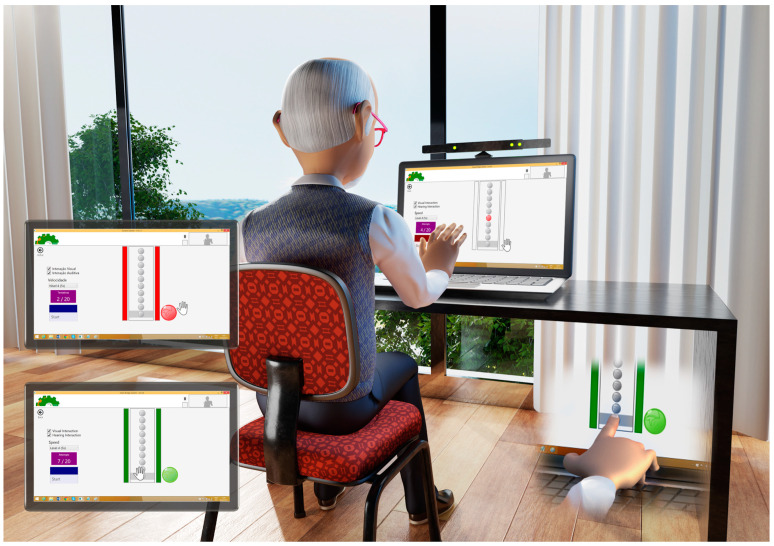
Task execution using virtual interface (Kinect—top right) and real interface (touch screen—bottom right). On the left is the demonstration of the feedback—Hit (green light—bottom left) and Miss (red light—top left).

**Figure 2 ijerph-20-03301-f002:**
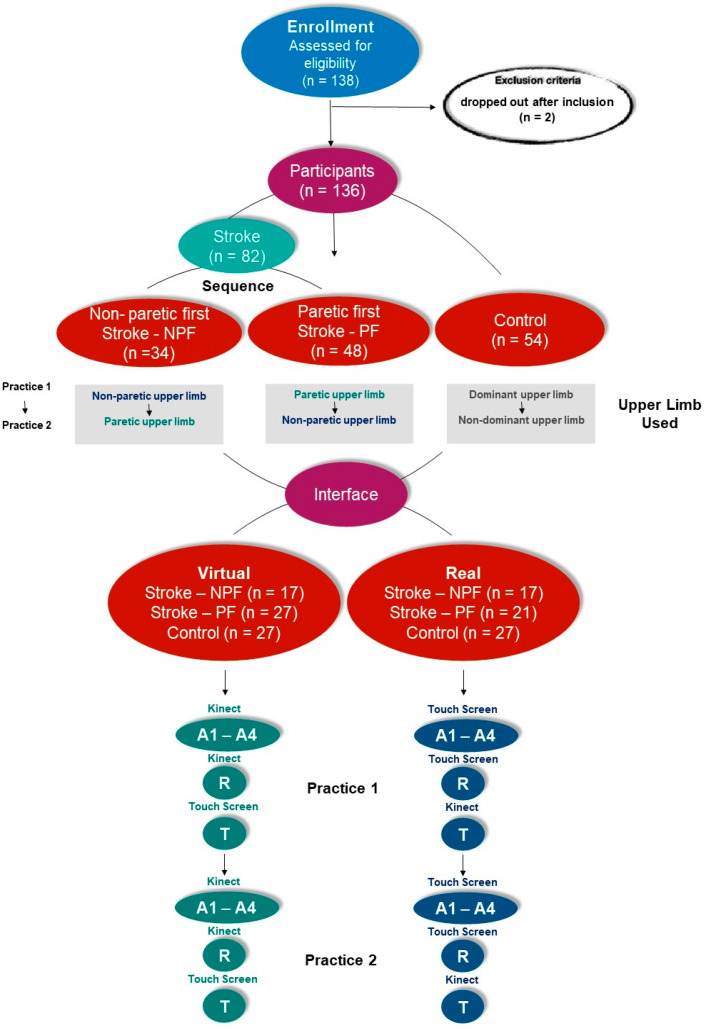
Flowchart of protocol sequence and groups. NPF: Non-paretic first; PF: Paretic first; A1–A4: Acquisition blocks; R: Retention block; T: Transfer block.

**Figure 3 ijerph-20-03301-f003:**
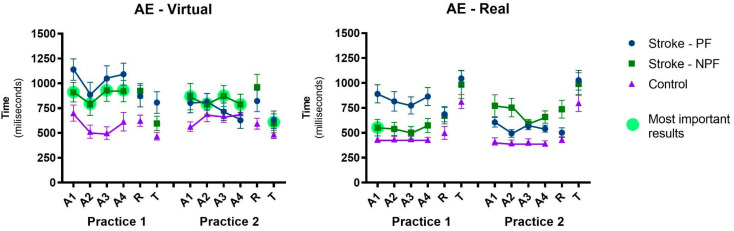
Representation of mean and standard error of the absolute error (AE) pattern for blocks, sequences, and groups on interfaces in all phases of the protocol, for stroke and control groups. AE: absolute error; A1: first acquisition block; A2: second acquisition block; A3: third acquisition block; A4: fourth acquisition block; R: Retention test block; T: Transfer test block—changing device; PF: paretic first; NPF: non-paretic first. ● Most important result (green sphere): only in the virtual interface, the participants with stroke that practiced with non-paretic limb first (Practice 1) showed better performance in the second sequence with paretic limb (Practice 2), and this group presented better results in the transfer to the real interface (Transfer, Practice 2). This can be reinforced by the comparison between the result of this transfer with the paretic upper limb group on real interface (i.e., Transfer of Practice 2 versus A1 of Practice 1—real interface).

**Figure 4 ijerph-20-03301-f004:**
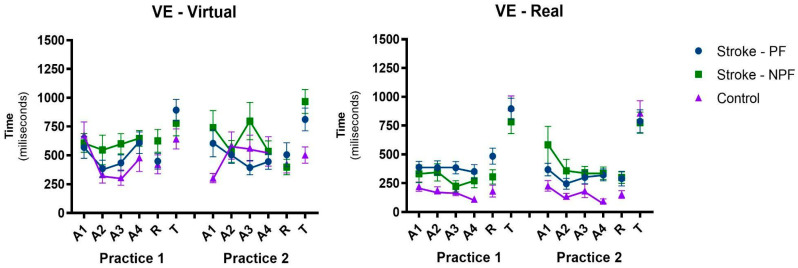
Representation of mean and standard error of the variable error (AE) pattern for blocks, sequences, and groups on interfaces in all phases of the protocol, for stroke and control groups. VE: variable error; A1: first Acquisition block; A2: second Acquisition block; A3: third Acquisition block; A4: fourth Acquisition block; R: Retention test block; T: Transfer test block—changing device; PF: paretic first; NPF: non-paretic first.

**Table 1 ijerph-20-03301-t001:** Sample characterization.

	Stroke—NPF (n = 34)	Stroke—PF (n = 48)	Control (n = 54)	*p*-Value
	Mean (SD) [CI 95%]	Mean (SD) [CI 95%]	Mean (SD) [CI 95%]	
Age (years)	56.2 (14.1) [51, 61]	59.0 (11.6) [56, 62]	57.4 (12.8) [54, 61]	0.603
OPS	2.24 (0.6) [2.05, 2.47]	2.09 (0.38) [1.98, 2.20]	-	0.070
FMA	53.3 (12.8) [48.8, 57.5]	54.4 (11.9) [50.7, 57.4]	-	0.156
MMSE	22.2 (5.2) [20.2, 23.9]	23.6 (4.6) [22.2, 24.8]	-	0.275
TUG	23.1 (11.9) [19.2, 27.4]	24.0 (12.5) [20.6, 28.0]	-	0.741
Lesion time (months)	31.5 (50.5) [16.6, 52.4]	41.5 (52.1) [26.9, 58.3]	-	0.330
AMAT FS	4.1 (1.4) [3.5, 4.5]	4.0 (1.4) [3.6, 4.4]	-	0.663
AMAT MQ	4.1 (1.4) [3.5, 4.5]	4.0 (1.5) [3.5, 4.4]	-	0.714
	n (%)	n (%)		
Sex				
Male	22 (28)	28 (34)	26 (38)	0.731
Female	12 (22)	20 (37)	28 (41)	
Type of Stroke				
Ischemic	25 (70)	41 (30)	-	0.088
Hemorrhagic	9 (85)	7 (15)	-	
Side of hemiparesis				
Right	17 (50)	23 (50)	-	0.441
Left	17 (54)	25 (46)	-	

NPF: Non-paretic first; PF: Paretic first; SD: Standard deviation; CI: Confidence interval; OPS: Orpington Prognostic Scale; FMA: Fugl-Meyer Assessment; MMSE: Mini-Mental State Examination; TUG: Timed Up and Go; AMAT: Upper Limb Motor Ability Test; FS: Functional Skills; MQ: Movement Quality.

## Data Availability

The data presented in this study are available on request from the corresponding author. The data are not publicly available due to other studies in progress.

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
