# Peer review of "Bilateral Transfer of Performance between Real and Non-Immersive Virtual Environments in Post-Stroke Individuals: A Cross-Sectional Study"

_ijerph, 2023, doi:10.3390/ijerph20043301_

Round 1

Reviewer 1 Report

In this study a group of post-stroke patients with upper limb paresis and and a group of controls underwent a protocol consisting in a repeated motor task which can be alternatively performed with a virtual (Kinect) or a real device 26 (Touchscreen). To explore the possibility of a bilateral transference the patients were divided in two groups: one performed the task with the non-paretic upper limb and then with the paretic, the other group performed the task with the paretic upper limb and then with the non-paretic. Results were analyzed to verify the following hypothesis: 1) the control group has better performance than patients in all protocol phases and all groups improve performance independent of the upper limb used, 2) all groups improve performance independent of the interface used, 3) both patients and control show transfer between upper limbs.

In my opinion the following corrections are necessary:

-1 Controls are not characterized in an exhaustive way: have they any pathological condition? Where and how have they been selected? A better characterization of controls is mandatory. Without this information we do not know which population they represent and comparation may be affected by biases.

- 2 Cognitive function of the individuals has been poorly explored (only MMSE). Neglected differences between groups regarding cognitive performance can be an important source of bias. This limitation should be signaled in the discussion.

- 3 Moreover neither markers of cognitive function nor motor or other clinical scores have been reported in Tab.1 for controls. May you present the scores of these individuals? If it was not collected this should be signaled as a limitation.

- 4 Regarding line 208 “five more attempts in Transfer (T), in this case the participants changed the interactive device”, may you explain in a more detailed way the reason of changing the interactive device during T phase? It is not totally clear.

- 5 The author declares some hypothesis, which were partially confirmed, but the primary aims of the study are not clear. Bilateral transference is a known event, usefulness of virtual reality in stroke rehabilitation is already known. If the author’s purpose is to prove the usefulness of virtual reality combined with bilateral transference in rehabilitation of post-stroke upper limb paresis it should be declared in a straighter mode (eventually in the hypothesis). In this case results with virtual and real devices should be explicitly compared.

- 6 In figure 3 and 4 do box-plots report means and standard deviations of AEs and VEs? Or other values? it must be explicitly specified in the captions.

- 7 Did you observed any adverse event related to virtual reality in the sample (like the one cited in the introduction – line 52)? It should be specified.

- 8 No limitations were described. At least moderate sample size and limited external validity, being a single center study, should be reported as limitations.

Reviewer 2 Report

The experimentation is well done. As always in this type of research, it will be better with more subjects.

It would be nice to have videos of the exercises, so the reader can have a better idea of the experiments done.

I would like a "future work" paragraph or section. It seems like you want to continue using the data, so it should be easy to give some general ideas of your research lines.

There are some English mistakes, like in page 13 "For those with chronic stroke and have difficulty accessing..." I suppose "have" is "that".

s always in this type of research, it will be better with more subjects.
